# Co-stimulation with opposing macrophage polarization cues leads to orthogonal secretion programs in individual cells

Andrés R. Muñoz-Rojas [1,5], Ilana Kelsey [1], Jenna L. Pappalardo[2], Meibin Chen[1] & Kathryn Miller-Jensen [1,3,4✉]

Macrophages are innate immune cells that contribute to fighting infections, tissue repair, and maintaining tissue homeostasis. To enable such functional diversity, macrophages resolve potentially conflicting cues in the microenvironment via mechanisms that are unclear. Here, we use single-cell RNA sequencing to explore how individual macrophages respond when co-stimulated with inflammatory stimuli LPS and IFN-γ and the resolving cytokine IL-4. These co-stimulated macrophages display a distinct global transcriptional program. However, variable negative cross-regulation between some LPS + IFN-γ-specific and IL-4-specific genes results in cell-to-cell heterogeneity in transcription. Interestingly, negative cross-regulation leads to mutually exclusive expression of the T-cell-polarizing cytokine genes *Il6* and *Il12b* versus the IL-4-associated factors *Arg1* and *Chil3* in single co-stimulated macrophages, and single-cell secretion measurements show that these specialized functions are maintained for at least 48 h. This study suggests that increasing functional diversity in the population is one strategy macrophages use to respond to conflicting environmental cues.

[1] Department of Biomedical Engineering, Yale University, New Haven, CT, USA. [2] Department of Immunobiology, Yale University School of Medicine, New Haven, CT, USA. [3] Department of Molecular, Cellular, and Developmental Biology, Yale University, New Haven, CT, USA. [4] Systems Biology Institute, Yale University, New Haven, CT, USA. [5] Present address: Department of Immunology, Harvard Medical School, Boston, MA, USA. ✉email: kathryn.miller-jensen@yale.edu

Macrophages respond to a large range of stimuli to aid in development, tissue repair, and immunity[1,2]. These polarization responses must be strong enough to defend against pathogens and tissue stress, but sufficiently plastic to accommodate changes in the microenvironment. The M1 inflammatory program, induced as a response to infectious stimuli (e.g., lipopolysaccharide [LPS] and/or interferon-gamma [IFN-γ]) and the M2 program, induced by resolving stimuli (e.g., interleukin-4 [IL-4]), represent two extremes on a spectrum of macrophage responses[3–5]. The M1 polarization program is typically associated with a proinflammatory and anti-bacterial phenotype, while the M2 program is associated with wound healing, tissue repair, and helminth response[5,6]. While this M1–M2 paradigm has been useful in uncovering key regulatory elements in the innate immune response, it is clear that macrophages in vivo display a more complex polarization response[7,8].

There are several possible sources for the complex polarization states observed in vivo. First, the vast number of stimuli that can activate macrophages enables these cells to display a large range of functional responses[7,9]. Second, because macrophage polarization does not induce terminal differentiation programs, functional responses by macrophages are plastic and may switch states in response to changing environmental conditions[10]. Additionally, macrophages in vivo are constantly exposed to multiple and sometimes conflicting cues. Examples of this include co-existing inflammatory and immunosuppressive signals during the resolution phase of inflammation[11] or the complex pro- and anti-tumor microenvironments established inside a tumor[12,13]. Together, the flexible and diverse range of macrophage responses and the stimuli they receive result in a complex polarization spectrum observed in vivo.

There is extensive crosstalk in the polarization programs in populations of macrophages presented with opposing cues[14,15]; however, the complexity of macrophage polarization has not yet been explored at single-cell resolution in vitro. Such single-cell measurements are important because macrophage populations display significant cell-to-cell heterogeneity in their responses even following acute stimulation with LPS[16–18]. The heterogeneity observed in vitro is also reflected in vivo in numerous studies of macrophage heterogeneity in a variety of disease states, including cancer, liver cirrhosis, and wound healing[19–23]. It is therefore important to characterize how individual macrophages respond to and resolve conflicting cues to coordinate a cohesive immune response within complex tissue microenvironments.

Here, we profile macrophages stimulated with LPS+IFN-γ, IL-4, or both by single-cell RNA sequencing (scRNA-seq) and by single-cell secretion profiling to explore how individual macrophage respond to opposing polarization cues presented simultaneously. On a global scale, mouse bone marrow-derived macrophages (BMDMs) stimulated with LPS+IFN-γ and IL-4 acquire a unique transcriptional state distinct from the states induced by single stimuli. However, extensive crosstalk occurs between polarization programs in individual macrophages presented with opposing cues, and the extent of this crosstalk varies across cells. Using a combination of neural networks and statistical analysis, we find a subset of genes from each single-stimulus gene program that are not expressed together in co-stimulated cells, including the T-cell-polarizing cytokines Il6 and Il12b, induced by LPS+IFN-γ, and the canonical M2-associated targets Arg1 and Chil3, induced by IL-4. Measurement of high-dimensional single-cell secretion profiles confirms that, in most cases, single cells express either an LPS+IFN-γ-like or IL-4-like secretion phenotype after long-term co-stimulation with opposing cues. Together, our results provide insight into the heterogeneous response of macrophages exposed to opposing polarization cues.

## Results

**Co-stimulated macrophages display a mixed gene expression program.** To test how co-stimulation with opposing polarization cues affects the gene expression programs of individual macrophages, we stimulated BMDMs with LPS+IFN-γ, IL-4, or a combination of LPS+IFN-γ and IL-4, and profiled the cells using scRNA-seq (Fig. 1a). In cell populations, we observed that in response to 10 ng/ml LPS + 10 ng/ml IFN-γ, expression of canonical M1-associated genes (including Nos2, Tnf, Il6, and Il12b) peaked between 2 and 6 h, while in response to 100 ng/ml IL-4, expression of canonical M2-associated genes (including Arg1 and Chil3) continued to rise by 8 h (Supplementary Fig. 1a). Therefore, we chose 6 h as an optimal stimulation time to capture both transcription programs.

We also explored a range of LPS and IL-4 dose combinations and observed that co-stimulation with 10 ng/ml LPS + 10 ng/ml IFN-γ and 100 ng/ml IL-4 resulted in significant cross-inhibition of LPS+IFN-γ-stimulated IL-6 and IL-12p40 and of IL-4-stimulated Arg1 and Chil3l3 at 24 h (Supplementary Fig. 1b). In contrast, LPS+IFN-γ-stimulated TNF and Nos2 were not significantly inhibited at any dose combination. Thus, we concluded that this dose combination might reveal interesting behaviors in individual cells.

For sc-RNAseq, we profiled ~1500 single cells per condition to ensure high-quality data with low doublet rates and high sequencing depth per cell (see "Methods" section). Our final data set had an average of 30,262 unique reads per cell and 4076 genes detected per cell. Stimulation with LPS+IFN-γ upregulated more genes than IL-4 stimulation when compared to unstimulated cells (1141 and 380 genes, respectively), with only 95 of these genes being induced by both conditions (Fig. 1b and Supplementary Data 1). Interestingly, both LPS+IFN-γ and IL-4 stimulation caused the apparent transcriptional downregulation of thousands of genes normally expressed in unstimulated cells, with the number of downregulated genes substantially outnumbering the number of upregulated genes (Fig. 1b).

We performed dimensionality reduction using uniform manifold approximation and projection (UMAP)[24,25] on the full transcriptional signature to visualize how cells mapped across treatments. We observed that almost all co-stimulated macrophages clustered separately from those stimulated with only one cue and from the unstimulated control population, suggesting that co-stimulation induced a distinct global transcriptional state (Fig. 1c). This result also demonstrated that most macrophages were able to respond to both stimuli. This is consistent with the observation that macrophages co-stimulated with LPS+IFN-γ and IL-4 displayed robust Stat1 and Stat6 phosphorylation downstream of the IFN-γ-receptor or IL-4-receptor, respectively, indicating that a majority of macrophages respond to both signals (Supplementary Fig. 1c).

Although the transcriptional state of co-stimulated macrophages was distinct from cells treated with LPS+IFN-γ or IL-4 alone, we observed cell separation within the co-stimulation cluster. For example, we observed heterogeneous expression of canonical genes associated with LPS+IFN-γ stimulation, Nos2 and Il12b, versus canonical genes associated with IL-4 stimulation, Arg1 and Mrc1 (Fig. 1d), suggesting that there is cell-to-cell variability in the extent of each individual cell's response to LPS+IFN-γ versus IL-4 stimulation. To further explore this possibility, we measured the Spearman correlation between genes uniquely upregulated by either LPS+IFN-γ or IL-4 alone across all single cells to identify which genes are co-expressed or mutually inhibited across single cells. As expected, genes upregulated by LPS+IFN-γ alone were more likely to exhibit positive correlations with other LPS+IFN-γ-induced genes, and either no correlation or negative correlation with IL-4-induced

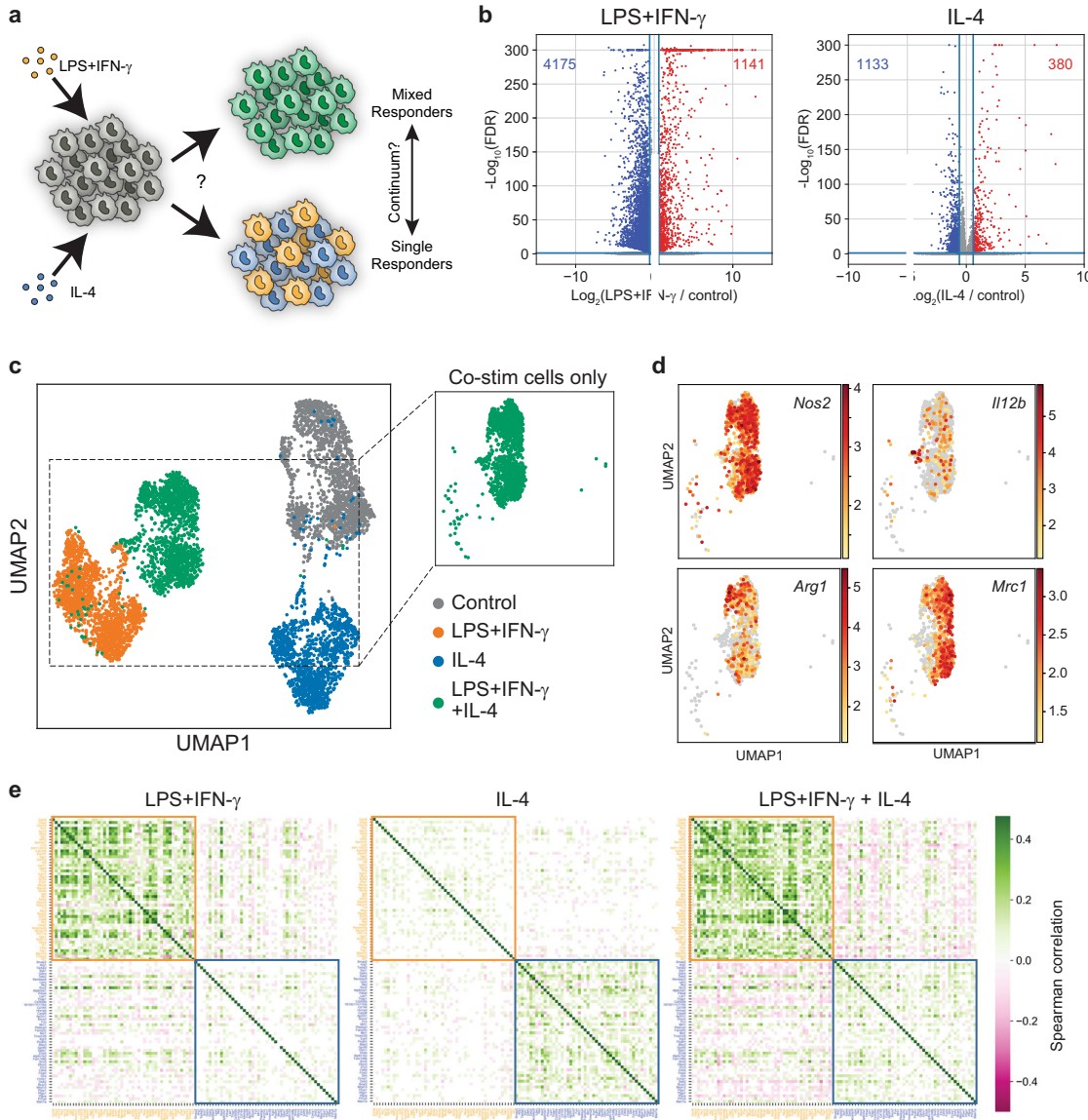

**Fig. 1 Co-stimulation induces a global mixed gene expression program across individual macrophages. a** Schematic depicting how co-stimulation could lead to either mixed responses or specialized responses in individual macrophages. **b** Volcano plot of differential gene expression after stimulation for 6 h with 10 ng/ml LPS + 10 ng/ml IFN-γ (left) or 100 ng/ml IL-4 (right) relative to control. Genes significantly upregulated (red) and downregulated (blue) are identified by a change in expression ≥ 1.5-fold and a false discovery rate (FDR) < 0.05 relative to expression in the untreated condition. FDR calculated by a Wilcoxon rank-sum test with Benjamini–Hochberg correction. **c**, **d** UMAP visualization of single cells from all samples colored by treatment (**c**) or expression intensity for canonical markers of interest (**d**). Color bar indicates gene expression levels, shown as ln(transcript count + 1). **e** Heatmaps of Spearman correlations for the top 50 unique upregulated genes following stimulation with LPS+IFN-γ, IL-4, or both, within single cells from each treatment condition. The top 50 genes upregulated by LPS+IFN-γ or IL-4 are highlighted and boxed in yellow and blue, respectively. The color bar indicates the sign and magnitude of the correlation coefficient. The coefficients for gene pairs with a correlation *p*-value > 0.05 are set to 0.

genes (Fig. 1e). Similarly, genes upregulated by IL-4 alone were more likely to be positively correlated with other IL-4-induced genes, and uncorrelated or negatively correlated with LPS+IFN-γ-induced genes. Co-stimulated cells generally exhibited weak positive correlations within the LPS+IFN-γ-induced genes and the IL-4-induced genes, while also exhibiting weak negative correlations between some genes across the two programs (Fig. 1e and Supplementary Fig. 1d). This suggests that on average macrophages expressing core genes of one program exhibited slightly reduced expression of core genes of the other program. Of note, not all correlations between LPS+IFN-γ-induced genes and IL-4-induced genes were negative, indicating that some LPS+IFN-γ-induced genes are co-expressed with IL-4-induced genes.

However, the strongest negative correlations were between genes induced by the two opposing stimuli, suggesting that single cells may skew gene expression towards either the LPS+IFN-γ or the IL-4 transcriptional program in response to co-stimulation.

**Co-stimulation induces heterogeneous transcriptional cross-regulation.** We next focused on how co-stimulation with LPS+IFN-γ and IL-4 affected the expression of genes uniquely upregulated by either LPS+IFN-γ or IL-4 alone (Fig. 2a), which we refer to as the core gene programs. For both LPS+IFN-γ- and IL-4-induced genes, co-stimulation with the other cue caused both transcriptional upregulation and inhibition in a subset of core genes belonging to each program, consistent with our own

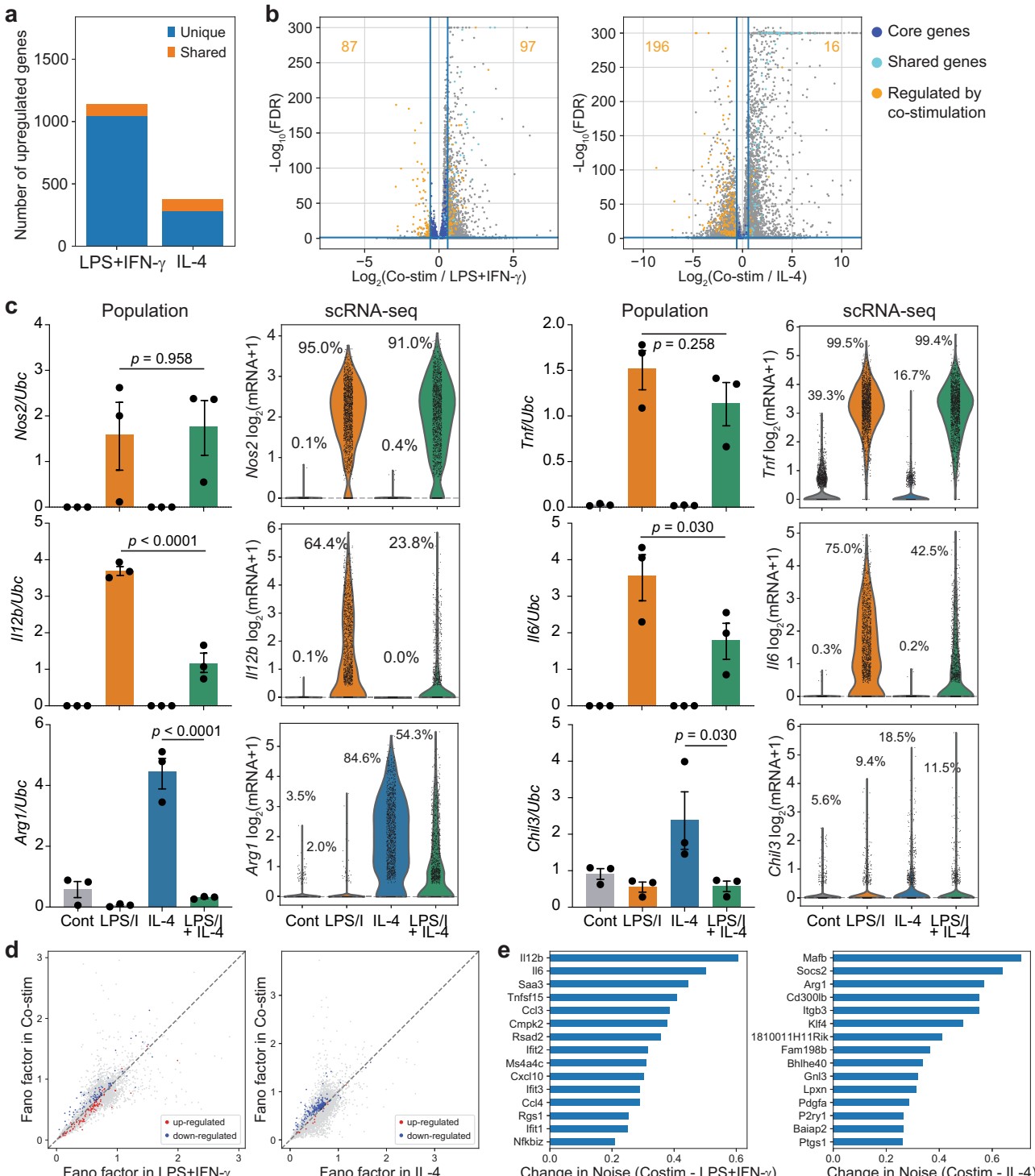

**Fig. 2 Cross-inhibition of gene expression after co-stimulation is not uniform across single cells. a** Number of unique (blue) and shared (yellow) genes upregulated after stimulation (6 h, 10 ng/ml LPS + 10 ng/ml IFN-γ or 100 ng/ml IL-4). **b** Volcano plot of differential gene expression after co-stimulation (6 h, 10 ng/ml LPS + 10 ng/ml IFN-γ and 100 ng/ml IL-4) relative to LPS + IFN-γ alone (left) and IL-4 alone (right). Dark blue dots (see key) indicate the unique core genes (UCGs) selectively induced by LPS + IFN-γ or IL-4, respectively. Cyan dots indicate shared genes induced by both LPS + IFN-γ or IL-4. Yellow dots indicate UCGs modulated by co-stimulation, identified by an FDR ≤ 0.05 and change in expression ≥ 1.5-fold relative to expression after stimulation with LPS+IFN-γ or IL-4 alone. FDR determined as in Fig. 1. **c** Transcript levels for indicated targets in BMDMs after stimulation for 6 h with media alone, LPS+IFN-γ, IL-4, or both, measured by population RT-qPCR (left) and scRNA-seq (right). Population mRNA levels are presented relative to those of the control gene *Ubc* (mean ± SEM, n = 3 biological replicates). Two-sided one-way ANOVA with Sidak correction for multiple comparisons. Single-cell data is presented as the ln(transcript count +1) from a single experiment. **d** Noise in gene expression (calculated by Fano factor) across single cells stimulated with LPS+IFN-γ versus co-stimulated cells (left) or IL-4 vs co-stimulated cells (right) for genes negatively (blue) and positively (red) regulated by co-stimulation. **e** Change in cell-to-cell gene expression noise of LPS+IFN-γ-induced genes (left) and IL-4-induced genes (right) in co-stimulated cells relative to single stimulation. Source data are provided as a Source Data file.

and previously reported observations[14]. Among the LPS+IFN-γ-induced core genes, 87 were inhibited by co-stimulation and 97 were augmented by co-stimulation, while among the IL-4-induced core genes, 196 were inhibited and 16 were augmented by co-stimulation (Fig. 2b).

We compared observations from our single-cell data set to population-level RT-qPCR data on co-stimulated cells for canonical LPS+IFN-γ and IL-4-induced core genes. Population-level measurements confirmed our scRNA-seq results, with LPS+IFN-γ-induced genes *Il12b* and *Il6* and IL-4-induced genes *Arg1* and *Chil3* exhibiting sensitivity to cross-inhibition, while other LPS+IFN-γ-induced genes, including *Nos2* and *Tnf*, were not sensitive to co-stimulation (Fig. 2c, bar plots). Inhibition or resistance to co-stimulation for this set of target genes was also observed at the protein level at 6 and 24 h (Supplementary Fig. 2a, b).

Importantly, although we observed substantial cross-inhibition in the cell population for some gene targets, the scRNA-seq data revealed that there was substantial cell-to-cell heterogeneity, such that some individual cells exhibited transcript levels that appeared uninhibited even after stimulation with both cues (Fig. 2c, violin plots). Given the significant cell-to-cell heterogeneity in cross-inhibition of the gene programs, one hypothesis is that macrophages are diversifying along these negatively regulated pathways. To identify cross-inhibited genes that are most variably expressed, we calculated the noise in gene expression as measured by Fano factor (variance divided by mean) for the LPS+IFN-γ- and IL-4-induced genes in both single and co-stimulated cells. Interestingly, we observed a subset of inhibited genes that upon co-stimulation decrease their mean expression across the population while increasing gene expression noise (Fig. 2d and Supplementary Fig. 2c). Strikingly, 9 out of the top 15 genes induced by LPS+IFN-γ and exhibiting increased noise were secreted cytokines or chemokines, including the T-cell-polarizing inflammatory cytokines *Il6* and *Il12b* (Fig. 2e, left). Notably, this gene set also includes NF-κB inhibitor zeta (*Nfkbiz*), which is known to stimulate the transcription of a subset of inflammatory response genes including *Il6* and *Il12b*[26]. IL-4-induced genes exhibiting increased noise included canonical genes like *Arg1* and *Socs2*, as well as transcription factors associated with promoting anti-inflammatory phenotypes like *Mafb* and *Klf4* (Fig. 2e, right)[27,28]. Altogether, these results suggest extensive transcriptional cross-regulation between the LPS+IFN-γ and IL-4 core gene programs that are gene-specific and that vary substantially from cell to cell.

We next asked whether cross-inhibited genes were enriched for pathways regulating certain cellular functions. The set of cross-inhibited genes from each single-stimulus program was analyzed using Ingenuity Pathway Analysis (IPA) to determine what cell functions are most sensitive to inhibition by co-stimulation (Supplementary Fig. 2d). LPS+IFN-γ-induced genes that were inhibited by co-stimulation were significantly enriched for pathways involved in communication between innate and adaptive immune cells and regulation of cytokine production, consistent with the targets identified in the noise analysis. IL-4-induced genes subject to inhibition by co-stimulation were enriched for STAT3 and JAK2 in hormone-like cytokine signaling, although the results were less significant than for LPS+IFN-γ. Altogether, the IPA result suggested that important functions typically stimulated by LPS+IFN-γ or IL-4, including secretion of cytokines and chemokines and STAT3 signaling, are inhibited by co-stimulation.

**Machine learning classifies some co-stimulated cells as single stimulus.** The extensive cell-to-cell heterogeneity and negative correlations between LPS+IFN-γ and IL-4 observed in single cells suggest that some macrophages might selectively express sets of LPS+IFN-γ-induced or IL-4-induced genes despite having been stimulated by both cues. To explore this hypothesis, we isolated the co-stimulated cells and performed dimensionality reduction using UMAP to visualize distinct clusters of cells after co-stimulation. We then calculated a combined gene-expression score for the unique core genes (UCGs) induced by LPS+IFN-γ or IL-4 for each single cell. When we overlaid these scores on the UMAP plot, we observed a clear separation between the cells that had a high LPS+IFN-γ UCG score and the cells that had a high IL-4 UCG score, suggesting selective expression of these gene sets by co-stimulated cells (Fig. 3a).

To further explore the expression state of co-stimulated cells, we next used a neural network (NN) to classify our co-stimulated cells into LPS+IFN-γ, IL-4, or mixed cell states. Specifically, we trained two separate classifiers on expression data from cells stimulated with LPS+IFN-γ alone and IL-4 alone in order to identify these single-stimulation signatures (Fig. 3b). We then used both classifiers to predict the cell states of our co-stimulated cells, which could be classified as LPS+IFN-γ-dominant, IL-4-dominant, mixed (i.e., positive prediction by both classifiers), or unclassified (i.e., negative prediction by both classifiers). We first trained the NN classifiers using the full transcriptome. In this case, the NN predicted that most cells were dominated by the LPS+IFN-γ gene program, with a few cells exhibiting a mixed state and almost no cells exhibiting an IL-4-dominant signature (Fig. 3c). However, when we restricted our training set to UCGs cross-inhibited with co-stimulation and trained a new set of NN classifiers, the number of co-stimulated cells classified as exhibiting a mixed state was reduced and a fraction of cells exhibiting an IL-4-dominant state emerged (Fig. 3c). In fact, the fractions of co-stimulated cells classified as LPS+IFN-γ-dominant or IL-4-dominant were both substantially larger than the mixed and unclassified fractions. When plotting the classification results back onto the UMAP representation, the cells classified as LPS+IFN-γ-dominant or IL-4-dominant were the same cells that had high LPS+IFN-γ or IL-4 UCG scores, respectively (Supplementary Fig. 3). Together, these results suggest that a subset of co-stimulated macrophages behave as if responding to only one stimulus for negatively cross-regulated transcriptional programs.

**Il6 and Il12b are expressed orthogonally with Arg1 in single cells.** Negative cross-regulation has the potential to create specialized functions within a subset of macrophages. If cross-regulation between two genes is strong, then these genes would not be anti-correlated but rather would be orthogonally expressed (i.e., the presence of gene A would significantly reduce the probability of expressing gene B such that they would not be expressed together). This expression pattern may be exacerbated in scRNA-seq data for genes with low transcript numbers due to dropout[29]. To quantify orthogonal expression, we expressed the scRNA-seq data as binary data with a threshold of 2 detected transcripts (see "Methods" section) and calculated the odds ratio between all pairwise combinations of downregulated genes. The odds ratio is the ratio of the odds of expressing A in the presence of B to the odds of expressing A in the absence of B. If the log of the odds ratio is negative, then the presence of B decreases the likelihood of A and is more likely to exhibit orthogonal gene expression.

We found that genes exhibiting significant negative odds ratios were relatively few, and were more likely to be observed between core genes upregulated by LPS+IFN-γ versus IL-4 than between core genes of the same program (Fig. 4a). We noted that transcripts for the T-cell-polarizing cytokines *Il6* and *Il12b* had

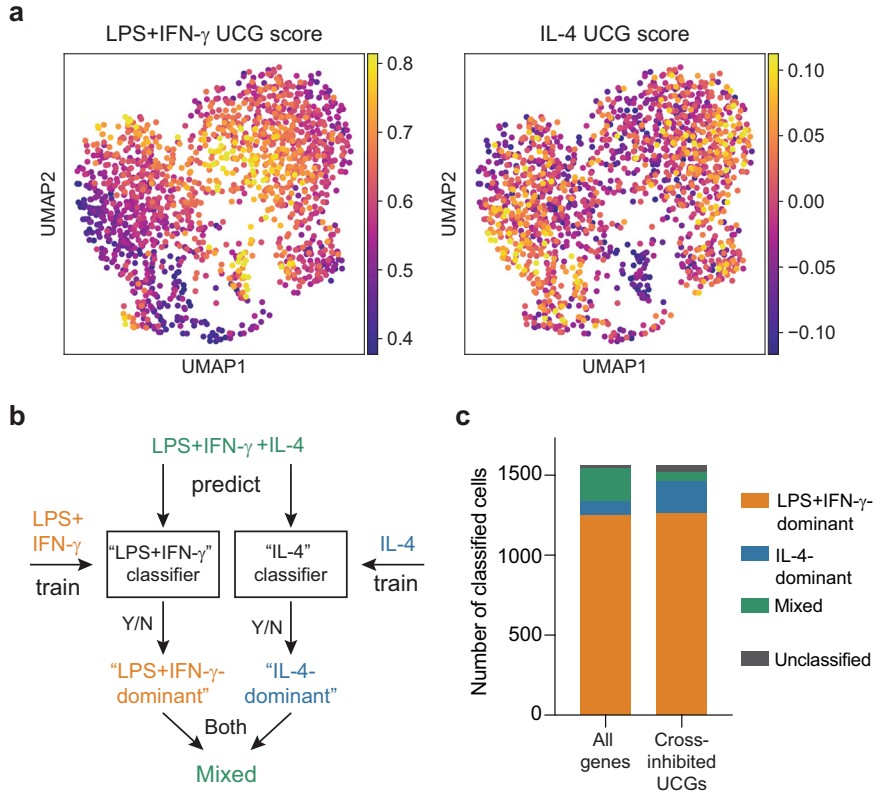

**Fig. 3 Neural network classifier suggests cross-inhibited genes are selectively expressed in single cells after co-stimulation. a** UMAP visualization of co-stimulated single cells, colored by the expression score of LPS+IFN-γ (left), or IL-4 (right) UCGs. **b** Schematic of the neural network classifiers built to categorize cells as LPS+IFN-γ-dominant, IL-4-dominant or Mixed. **c** Results of the neural network classification of the transcriptional state of co-stimulated cells. Bars represent the number of cells identified as LPS+IFN-γ-dominant (yellow); IL-4-dominant (blue); mixed (green); or unclassified (gray). The classifiers were trained on either all genes or UCGs that displayed cross-inhibition after co-stimulation. Source data are provided as a Source Data file.

negative odds ratios with *Arg1* (Fig. 4a). To confirm the orthogonal expression of these two pathways, we plotted the sc-RNAseq expression for *Il6* and *Il12b* against *Arg1* across all conditions (Fig. 4b). Plotting single-cell expression of these transcripts revealed a striking orthogonal expression pattern: cells with high levels of *Il6* or *Il12b* had no or very low expression of *Arg1*, and vice versa (Fig. 4b). Orthogonal expression with *Il12b* and *Il6* was also observed with the IL-4-associated target *Chil3* (Supplementary Fig. 4a). Thus, it appears that select LPS+IFN-γ and IL-4 targets are rarely expressed coincidentally in the same individual cell.

We next looked for transcription factors (TFs) related to these targets that might play a role in selective expression. We observed that *Nfkbiz*, a positive TF for the secondary response genes *Il6* and *Il12b*[19], also had a negative odds ratio with *Arg1*, suggesting that this pathway may be orthogonally regulated with the *Arg1* pathway (Fig. 4a). In support of this hypothesis, *Nfkbiz* had a negative odds ratio with *Klf4*, a positive TF for *Arg1* and *Chil3* that also negatively regulates LPS-stimulated proinflammatory genes (Fig. 4a, c)[28]. Importantly, we previously noted that *Nfkbiz* and *Klf4* are TFs that are subject to negative cross-regulation and increase their noise after co-stimulation (Fig. 2e). We then measured the expression of these TFs by RT-qPCR in a cell population and confirmed that *Nfkbiz* and *Klf4* are inhibited after co-stimulation (Fig. 4d, e). Together these results suggest that macrophages co-stimulated with LPS+IFN-γ and IL-4 may orthogonally express the TFs *Nfkbiz* and *Klf4*, resulting in orthogonal expression of *Arg1* and *Chil3* versus the secondary cytokine genes *Il6* and *Il12b*.

We did not observe orthogonal expression with *Arg1* or *Klf4* for transcripts encoding primary cytokines and chemokines including *Ccl5* and *Tnf* (Supplementary Fig. 4b). This is consistent with the fact that the expression of *Tnf* and other primary genes is *Nfkbiz*-independent[26,30]. Overall, our results strongly suggest that co-stimulated macrophages specialize in the production of transcripts for *Arg1* and *Chil3* versus *Il6* and *Il12b*, but not primary secreted cytokines/chemokines, at least partially through the heterogeneous regulation of *Klf4* and *Nfkbiz* expression.

**Orthogonal transcription leads to subsets with specialized secretion.** Although our results demonstrate orthogonal expression of *Klf4* and *Nfkbiz* transcripts and their targets *Arg1* and *Chil3* or *Il6* and *Il12b*, respectively, in LPS+IFN-γ and IL-4 co-stimulated macrophages at 6 h, it is unclear whether this specialization is sustained long enough to direct macrophages to have distinct functions. To explore this possibility, we used a multiplexed single-cell secretion assay that we previously used to measure the heterogeneity of LPS-stimulated macrophages[16,31]. We stimulated BMDMs cultured in the single-cell device with LPS+IFN-γ, IL-4, or a combination of LPS+IFN-γ+IL-4, and captured the secretion of 11 different proteins for 48 h to obtain an integrative measurement of the final secretion state for each macrophage. We measured a combination of inflammatory secreted proteins including TNF, CCL5, IL-6, and IL-12p40. Because Arg1 is not secreted, we measured Chi3l3 (the protein product of *Chil3*) as the primary secreted protein in response to IL-4 stimulation.

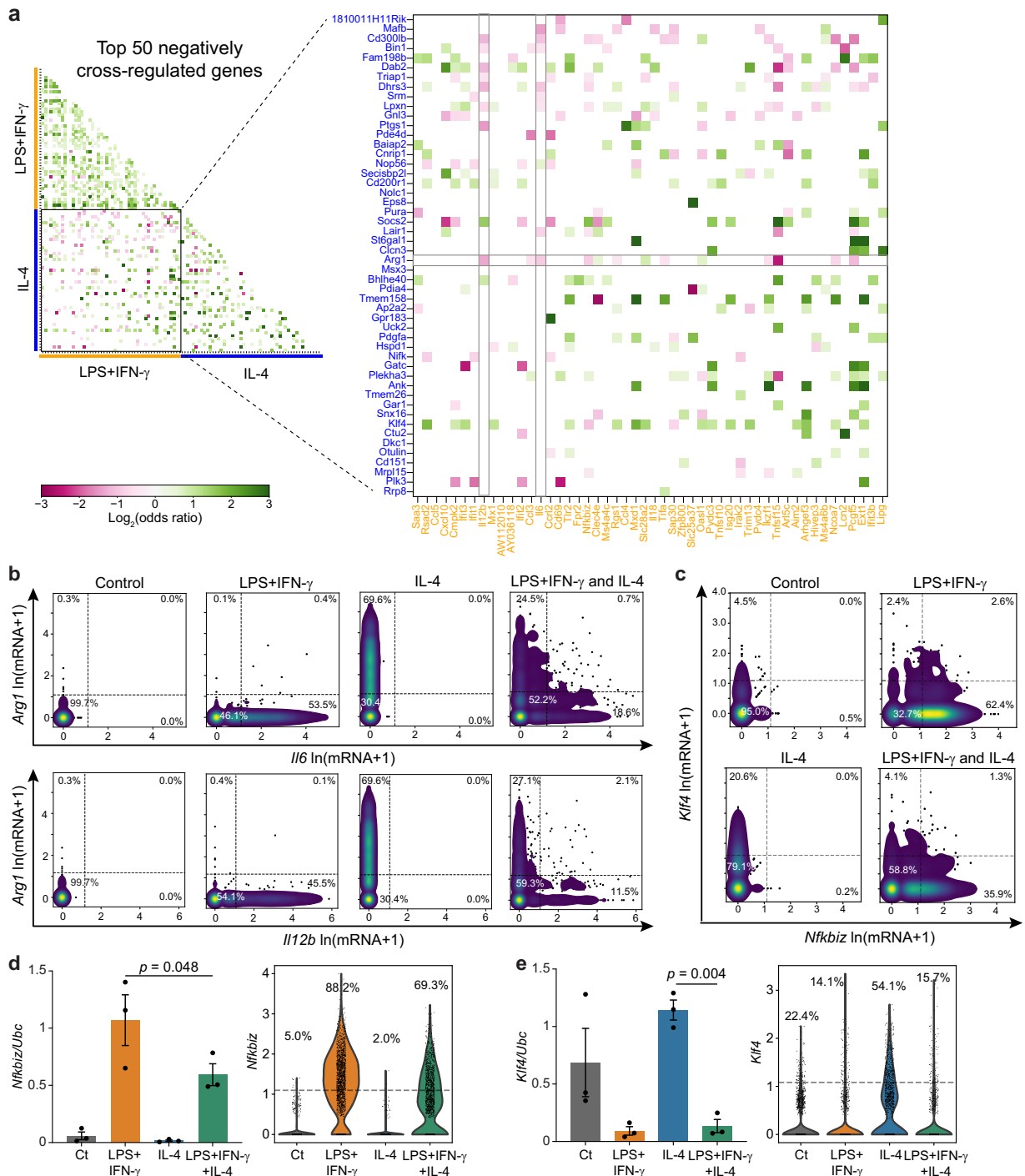

**Fig. 4 *Il6* and *Il12b* and their TF gene *Nfkbiz* are expressed orthogonally with *Arg1* and its TF gene *Klf4*. a** Pairwise odds ratios for the top 50 downregulated genes in the LPS+IFN-γ (yellow) and IL-4 (blue) core gene programs measured in co-stimulated cells. Data are presented as the log2(odds ratio). The color bar indicates the strength and magnitude of the association. Odds ratios with a *p*-value > 0.05 (determined by two-sided Fisher's exact test) are set to 0. **b, c** Density scatter plots of scRNA-seq transcript counts across individual cells for *Arg1* vs. *Il6* (**b**; top) and *Il12b* (**b**; bottom) and *Klf4* vs. *Nfkbiz* (**c**). Data are represented as ln(transcript count +1). **d, e** Population RT-qPCR measurements (left) and violin plots of scRNA-seq measurements (right) of *Nfkbiz* (**d**) and *Klf4* (**e**) after stimulation for 6 h with media alone, LPS+IFN-γ, IL-4, or both. Population mRNA levels are presented relative to those of the control gene *Ubc* (mean ± SEM, *n* = 3 biological replicates). Two-sided one-way ANOVA with Sidak correction for multiple comparisons. Single-cell data is presented as the ln(transcript count +1) from a single experiment. Source data are provided as a Source Data file.

After 48 h of co-stimulation, we observed orthogonal secretion of IL-6 and IL-12p40 and the IL-4-induced marker Chi3l3 (Fig. 5a), similar to what we observed at the transcript level at 6 h. Indeed, we found very few cells that co-secreted IL-6 and Chi3l3 or IL-12p40 and Chi3l3 (Fig. 5b, green). This orthogonal behavior

was not observed between CCL5 and Chi3l3 or TNF and Chi3l3 (Supplementary Fig. 5a), consistent with the scRNA-seq results (Supplementary Fig. 5b). To verify orthogonal secretion, we calculated the odds ratio between all pairwise combinations of secreted proteins in co-stimulated cells to measure whether the

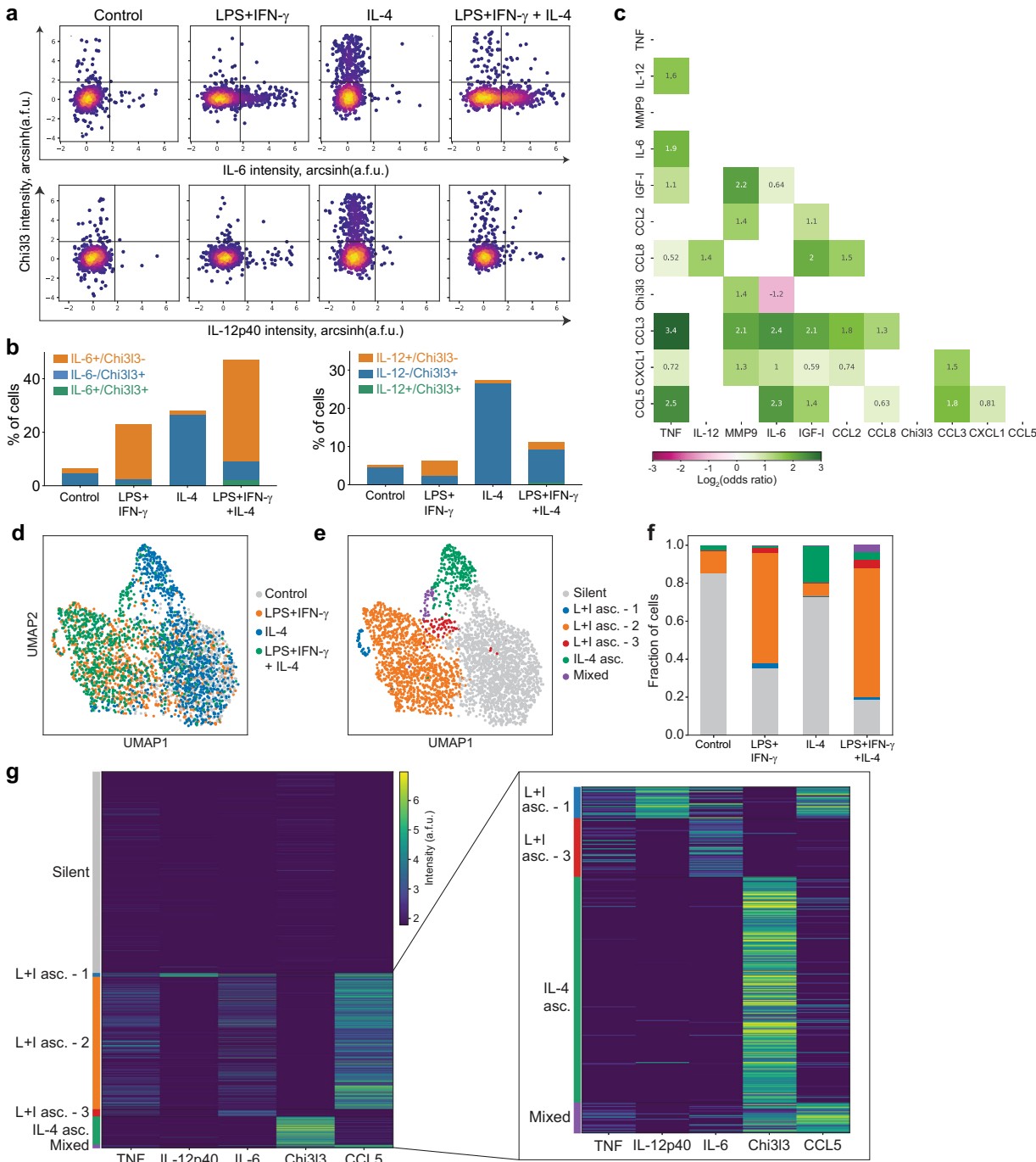

**Fig. 5 Single-cell secretion assay identifies long-term specialization in co-stimulated cells.** A multiplexed single-cell secretion assay was used to measure cytokine/chemokine production in individual BMDMs stimulated for 48 h with media alone, 10 ng/ml LPS + 10 ng/ml IFN-γ, 100 ng/ml IL-4, or both. **a** Density scatter plots of single-cell secretion intensity (a.f.u.) across individual cells for Chi3l3 vs IL-6 (top) and Chi3l3 vs IL-12p40 (bottom). **b** Quantification of single- and double-positive cells after co-stimulation for Chi3l3 and IL-12p40 (left) and Chi3l3 and IL-6 (right). **c** Pairwise odds ratios for all secreted proteins measured in co-stimulated cells. Data are presented as the log₂(odds ratio). The color bar indicates the direction and magnitude of the association. Odds ratios with an associated *p*-value > 0.05 (determined by two-sided Fisher's exact test) are set to 0. **d-e** UMAP visualization of single-cell secretion data for 5 proteins (Chi3l3, CCL5, IL-6, IL-12p40, and TNFα), colored by stimulation (**d**) or consensus cluster (**e**). **f** Quantification of the fraction of cells present in each consensus cluster identified by ensemble clustering. **g** Heatmap of secretion profiles for the five secreted proteins in **d** grouped by consensus cluster. Inset is a magnified version of the smaller clusters. All single-cell secretion data presented is pooled from two biological replicates. L + I asc. LPS + IFN-γ-associated; IL-4 asc. IL-4-associated. Source data are provided as a Source Data file.

secretion of a given protein is more, less, or as likely in a single cell when that cell is secreting the paired protein. We observed that the only pair of proteins with a negative log₂(odds ratio) was IL-6 and Chi3l3 (odds ratio = 0.44 and log₂(odds ratio) = −1.2),

indicating that if a cell is secreting IL-6, the odds of that cell also secreting Chi3l3 are 56% lower than the odds of secreting Chi3l3 when IL-6 is not being secreted (Fig. 5c). These data demonstrate that the specialization observed in transcription at 6 h is

maintained for secretion at 48 h, such that co-stimulated cells either secrete IL-6 and IL-12p40 or Chi3l3.

Finally, we explored the extent to which subsets of co-stimulated macrophages showed secretion programs skewed towards LPS+IFN-γ versus IL-4 stimulation. To test this, we performed dimensionality reduction and unsupervised ensemble clustering[32] on five proteins distinguishing LPS+IFN-γ- and IL-4-stimulated cells in the single-cell secretion data set (specifically: TNF, CCL5, IL-6, IL-12p40, and Chi3l3). We found that the cells stimulated with individual stimuli divided into two separate branches in the low-dimensional UMAP projection, while the co-stimulated cells spread across these two branches (Fig. 5d). Clustering analysis revealed that all conditions contained a cluster of silent cells with low secretion of all proteins (Fig. 5e–g). Additionally, cells stimulated with LPS+IFN-γ contained clusters of cells secreting a combination of TNF, IL-12p40, IL-6, and CCL5 (Fig. 5e–g). Cells stimulated with IL-4 were composed of mostly silent cells, as well as a distinct cluster secreting only Chi3l3 (Fig. 5e–g). Co-stimulated cells were composed of a distribution of the three LPS+IFN-γ-associated clusters and the IL-4-associated cluster, as well as a small mixed cluster with cells co-secreting Chi3l3, CCL5, and TNF (Fig. 5e–g). Overall, we conclude that for secretion, a majority of macrophages co-stimulated with LPS+IFN-γ and IL-4 exhibit secretion profiles consistent with only one of these stimuli.

## Discussion

Macrophages exhibit diverse polarization states in vivo in response to complex cues in their microenvironments, but our understanding of how individual macrophages respond to simultaneous cues is limited. In this study, we used a combination of population and single-cell measurements and computational analyses to explore how macrophages respond when simultaneously presented with LPS+IFN-γ and IL-4. We found that while co-stimulated macrophages displayed a distinct global transcriptional program, variable negative cross-regulation between some LPS+IFN-γ- and IL-4-stimulated gene programs resulted in significant cell-to-cell variability, such that some co-stimulated macrophages are skewed towards one of the two transcriptional programs. In particular, our results suggest that negative cross-regulation by the TFs Klf4 and Nfkbiz leads to the orthogonal expression of the secondary Th1 cytokines Il6 and Il12b with Arg1 and Chil3, and that this results in macrophages with specialized secretion functions that are maintained for at least 48 h.

To further explore this diversification, we used a combination of correlation analysis, neural network classifiers, and dimensionality reduction to examine the cell state of co-stimulated cells. We found that the polarization state identified by these analyses was dependent on the transcriptional program analyzed. When looking at a global transcriptional state, most cells are classified as mixed cells; however, when we looked at genes induced by LPS+IFN-γ that are susceptible to cross-inhibition, a subset of co-stimulated cells appeared more specialized. Indeed, we show that some of the most widely accepted markers for M1-like and M2-like polarization states can classify co-stimulated cells as specialized, whereas others would classify as mixed. Thus, in agreement with previous reports, our results argue that overall polarization is better understood as a spectrum; however, our results add the insight that negative cross-regulation significantly reduces the probability that individual macrophages exhibit an intermediate state in this spectrum for a subset of gene programs[3,4]. This gene set-dependent classification may partially explain the difficulty in assigning polarization states in vivo—indeed, many studies find

contradicting results when trying to identify the polarization state of many tissue-resident macrophages[33,34].

Our observation that the TFs Klf4 and Nfkbiz are orthogonally expressed suggests a regulatory mechanism for the specialization of IL-6 and IL-12p40 secretion versus Chi3l3 secretion. Klf4 is an example of a macrophage-specific TF that regulates specialized gene programs[35]. Based on sequence analysis, there is a binding site for Klf4 on the promoter of Nfkbiz[35], and thus it is possible that Klf4 directly negatively regulates Nfkbiz upon co-stimulation. Recent work by Piccolo et al. identified several transcriptional and epigenetic mechanisms that regulate the integration of IFN-γ and IL-4 signals and are responsible for their cross-regulation[14]. Furthermore, recent developments in single-cell epigenetic measurements have identified how cell-to-cell variation can be encoded via chromatin accessibility and can direct complex biological decisions like hematopoietic differentiation[33,34]. Future work focused on measuring differences in the accessibility of distinct promoters might explain the observed specialization and help to determine whether these partially specialized cells exist in a predisposed state before stimulation, or if the specialization is imposed and reinforced after receiving the stimuli.

Macrophage-secreted cytokines and chemokines are essential for coordinating the immune response in complex tissue microenvironments. We observed nearly orthogonal secretion of IL-4-stimulated Chi3l3 and LPS+IFN-γ-stimulated IL-6 and IL-12p40 (Fig. 5a), which are essential cytokines for inducing T-cell polarization[35]. The observed diversification of single macrophages across a range of secretion programs, when presented with opposing cues, may enable macrophages to more quickly adapt to changing environments. This concept of "bet-hedging" has been described in many multi-cellular systems as a way to increase the robustness of the population, including bacterial responses to resource availability as well as mammalian diversification of NF-κB signaling or T-cell activation[36–38]. It is possible that macrophages display this partial specialization as a way to execute the dynamic nature of immune responses, such as the quick transition observed during the resolution of inflammation, where macrophages have to coordinate a successful change from an anti-microbial to a tissue-repair phenotype. Indeed, the ability of macrophages to quickly adapt to changing environments has been exploited to design therapies that induce anti-tumor immunity in melanoma[19].

There are several limitations to our study. First, we chose to measure single-cell transcription at 6 h and at a single dose chosen to balance the response dynamics of LPS+IFN-γ and IL-4 and the extent of negative regulation upon co-stimulation (Supplementary Fig. 1a, b). However, shorter or longer times after stimulation and different dose combinations will likely reveal additional aspects of macrophage response heterogeneity, as has been previously demonstrated[39]. In addition, we differentiated monocytes with M-CSF, and we observed underlying heterogeneity in these control populations prior to subsequent stimulation (Fig. 1c, control). Other studies have differentiated monocytes with GM-CSF, and in these cases, the resulting cell populations were also complex and heterogeneous[40]. Thus, pre-existing heterogeneity resulting from macrophage differentiation will affect the observed responses to subsequent cues. Given these qualifications, our study represents only a starting point; many additional conditions will need to be explored to fully appreciate the heterogeneity in macrophage responses.

The single-cell response of BMDMs to co-stimulation with LPS+IFN-γ and IL-4 has important implications for understanding macrophage responses to complex cues that co-exist in vivo. Our results contribute to growing evidence that individual macrophages distinctly process combinatorial environmental cues[39,41]. In all these cases, variability in the intracellular networks was the

primary source of heterogeneity explored. However, other studies have identified the importance of paracrine and autocrine signals in regulating macrophage polarization and heterogeneity[16,42,43]. For example, induction of the classical M2-associated gene *Arg1* during mycobacterial infection is regulated by autocrine–paracrine signaling, suggesting that autocrine and paracrine signaling networks could further regulate mixed and specialized subpopulations of macrophages[44,45].

Finally, the various contributors to heterogeneity in vivo are not yet clear. Due to the limited diffusion distances and communication capacity of in vivo settings, it has become apparent that macrophage heterogeneity in the body is at least partially explained by the local microenvironment around individual macrophages[16,17]. For example, different locations within tumor microenvironments can induce functionally distinct macrophages that are important in regulating disease progression[19,46]. It is, therefore, possible that in vivo, some of the observed heterogeneity and specialization is location dependent. In addition to genetic and local regulation, recent work has highlighted how metabolic state and circadian rhythm have profound effects on macrophage functions[47–49]. It is possible that differences in metabolic states and molecular clocks help generate the diversity required to induce partially specialized macrophages after co-stimulation. Thus, determining the relative importance of non-genetic cell-to-cell heterogeneity vs. other in vivo factors will be an important future direction.

## Methods

**Mice and cell culture**. Male or female C57BL/6J mice 6–8 weeks of age were purchased from Jackson Laboratories. Mice were housed at room temperature according to the standard housing conditions of the Yale Animal Resources Center for one week before bone marrow extraction. Bone marrow-derived macrophages were generated as previously described[50]. Briefly, bone marrow was extracted from the hind legs of the mouse with a syringe. After red blood cell lysis with ammonium-chloride-potassium lysis buffer (Lonza), cells were incubated for 4 h at 37 °C with 5% $CO_2$ in a non-tissue culture (TC) treated plastic petri dish with BMDM media (RPMI supplemented with 10% FBS, 100 U/ml penicillin, 100 μg/ml streptomycin, 1% sodium pyruvate, 25 mM HEPES buffer, 2 mM L-glutamine, and 50 μM 2-mercaptoethanol). After 4 h, the non-adherent cells were transferred to a new petri dish and incubated with BMDM media + 20 ng/ml macrophage-colony stimulating factor (M-CSF; Peprotech). After 3 days, an additional 10 ml of BMDM media + 20 ng/ml M-CSF was added to the plate. 6 days after plating, cells were harvested in PBS + 5 mM EDTA with gentle scraping, and the cell suspension was used to seed new non-TC treated dishes or microwell devices. All mice were housed in the Yale Animal Resources Center in specific pathogen-free conditions. All animal experiments were performed according to the approved protocols of the Yale University Institutional Animal Care and Use Committee.

**In vitro BMDM experiments**. BMDMs were plated and stimulated in BMDM media + 10 ng/ml M-CSF. Cells were plated in non-TC treated 6- or 12-well plates (Falcon) at a density of 100,000 cells/cm$^2$ and allowed to adhere overnight. Cells were then stimulated at the indicated doses and times using LPS (Invivogen), IFN-γ (Peprotech), and/or IL-4 (Peprotech).

**Quantification of secretion in population**. To measure secretion in populations of cells, supernatant from stimulated BMDMs was collected and stored at 4 °C for no more than a week. Secreted protein levels were measured using enzyme-linked immunosorbent assay (ELISA) kits according to the manufacturer's recommendations. Each ELISA kit reference number and manufacturer are listed in Supplementary Table 1.

**RT-qPCR**. RT-qPCR was performed as previously described[51]. Briefly, RNA was extracted using the RNEasy Mini Kit (Qiagen). Genomic DNA was removed on-column with RNase-free DNase (Qiagen) or with the TURBO DNA-free kit (Ambion), and complementary DNA (cDNA) was synthesized using a dT oligo primer and Superscript III RT (Invitrogen). After dilution in nuclease-free water, cDNA was quantified using SYBR-green for quantitative reverse transcription-polymerase chain reaction on a CFX Connect Real-Time System (Bio-Rad) with the following amplification scheme: 95 °C denaturation for 1.5 min followed by 40 cycles of 95 °C denaturation for 10 s, 65 °C annealing for 10 s, and 72 °C elongation for 45 s with a fluorescence read at the end of the elongation step. This was followed by a 65–85 °C melt-curve analysis with 0.5 °C increments. All samples

were normalized to the house-keeping gene *Ubc* (ubiquitin). Primers are reported in Supplementary Table 2.

**Flow cytometry**. Macrophages were lifted with gentle scraping in ice-cold PBS + 5 mM EDTA. For intracellular protein staining, cells were blocked with Fc receptor (CD16/CD32) antibody (eBioscience, clone 93) at 1:200 dilution on ice for 15 min in FACS buffer (PBS + 2% FBS). Then the cells were fixed with Cytofix/Cytoperm (BD Biosciences) and stained in 50 μl with anti-NOS2-AlexaFluor488 at 1:500 dilution (eBioscience, clone CXNFT) and anti-ARG1-APC at 1:10 dilution (R&D, polyclonal) for an hour at 4 °C. For phospho-flow, cells were fixed immediately after lifting with PhosFlow Fix buffer for 10 min at 37 °C, and subsequently permeabilized with PhosFlow Perm Buffer III for 30 min on ice (BD Biosciences). Cell suspensions were then blocked with Fc receptor antibody as above, and stained with anti-pSTAT1(Y701)-AlexaFluor488 at 1:50 (Cell Signaling Technologies, clone 58D6) and anti-pSTAT6(Y641)-AlexaFluor647 at 1:50 (Cell Signaling Technologies, clone D8S9Y). All data were acquired on an Accuri B6 flow cytometer (BD Biosciences), and analyzed with FlowJo (FlowJo, LLC). The gating strategy is shown in Supplementary Fig. 6.

**Single-cell RNA sequencing**. Stimulated BMDMs were lifted in ice-cold PBS + 5 mM EDTA with gentle scraping, washed, counted, and immediately processed at low density for scRNA-seq using the 10× platform (10× Genomics). Library construction and sequencing were performed by the Yale Center for Genomic Analysis according to the manufacturer's recommendations.

**scRNA-seq analysis**. The sequencing data were processed using the standard cellranger pipeline (10× Genomics). Further downstream analysis was performed using the Python package scanpy[52]. Cells were filtered for quality control to avoid doublets and dead cells, and counts were normalized using the scran package in R in a standard processing pipeline previously described[53,54]. Dimensionality reduction and visualization were performed within the scanpy package. Correlation and odds ratio analysis were performed using custom scripts in Python. To find core genes for each polarization program, differential testing using Wilcoxon rank-sum test with Benjamini–Hochberg correction for multiple comparisons was performed to find differentially expressed genes between stimulated cells and control cells. Core genes were defined as genes with a minimum fold change of 1.5, maximum FDR of 0.05, and expressed in at least 15% of the stimulated cells. From this list of core genes, we found the UCGs that were only induced by a single stimulation, and not by both. The analysis described above relied on the following additional software packages: anndata2ri, rpy2, pandas, and seaborn.

**Microwell assay for single-cell secretion profiling**. The single-cell secretion profiling experiments were performed as previously described[10,22], with some modifications for the analysis of primary mouse BMDMs. In brief, the capture antibodies (Supplementary Table 1) were flow patterned onto epoxysilane-coated glass slides (SuperChip; ThermoFisher). The polydimethylsiloxane nanowell arrays and antibody barcode glass slides were blocked using complete BMDM media. Fully differentiated BMDMs were resuspended in complete BMDM media and 10 ng/ml M-CSF and supplemented with 125 nM of live cell marker (Calcein AM; ThermoFisher) to facilitate automatic live-cell detection. The cells were added to the device and allowed to adhere overnight. The next day, cells were stimulated and covered with the antibody barcode slide, secured with screws, and allowed to incubate for 48 h. At the end of the incubation period, the device was imaged with an automated inverted microscope (Eclipse Ti; Nikon or Axio Observer Z1; Zeiss) to record well position and cell locations. The device was then disassembled to perform the sandwich immunoassay. The glass slide was incubated with a mixture of detection antibodies (Supplementary Table 1) for 1 h, followed by incubation with 20 μg/ml streptavidin-APC (eBioscience) for 30 min, rinsed with PBS and water, and finally scanned with a Genepix 4200A scanner (Molecular Devices).

**Single-cell secretion profiling data processing**. Device images were analyzed using a custom script in MATLAB (MathWorks) to automatically detect well location and number of cells per well, extract all signals from each well, and process the data (https://github.com/Miller-JensenLab/Single-Cell-Analysis). In brief, after automatic well and live-cell detection, signal image registration, and manual curation, the software automatically extracted the intensity signal from each antibody for all the nanowells in the device. The signal across the chip for each individual antibody was normalized by subtracting a moving Gaussian curve fitted to the local zero-cell well intensity levels. A secretion threshold for each antibody was then set at the 99th percentile of all normalized zero-cell wells. Finally, the data were transformed using the inverse hyperbolic sine with a cofactor set at 0.8× secretion threshold. To further visualize the data, custom Python scripts were used to generate UMAP visualizations and density scatter plots. Odds ratios were calculated using the same methods as for scRNA-seq data.

**Neural network classifier**. The neural network classifier was built using the machine learning python package scikit-learn[55]. The data from the cells receiving a single stimulus was split into a training and a testing data set to build and test the

classifier. A one-vs-the-rest (OvR) multilabel classifier strategy was used to enable non-mutually exclusive labels and identify mixed cells. Briefly, two multi-layer perceptron (MLP) classifiers were trained on the LPS+IFN-γ$^-$ and IL-4-stimulated cells to distinguish cells stimulated with each cue. The MLPs had three hidden layers and used the hyperbolic tan function as its activation function. Then the OvR multilabel classifier was used to predict the identity of co-stimulated cells, which labeled each cell as LPS+IFN-γ-stimulated, IL-4-stimulated, both, or none (no prediction). The accuracy of each classifier was >99% and was tested using cross-validation with the single-stimuli test data.

**Ensemble clustering**. Ensemble clustering was performed with the openensembles python package from the Naegle lab (https://github.com/NaegleLab/OpenEnsembles)[32,56]. Briefly, clustering was performed by sweeping across multiple clustering algorithms (Affinity Propagation, agglomerative, spectral, Birch, kmeans), across several distance and linkage metrics (average, complete, euclidean, cosine, ward, l1, and l2) and across a wide range of k-values (that determines the number of clusters to identify). A consensus clustering solution was found by linking the co-occurrence matrix[32,56]. Clusters with less than three cells were discarded for analysis.

**Statistics**. Data were presented as mean ± SEM unless otherwise specified. Statistical analysis was generally performed by two-sided, unpaired Student's $t$-test or one-way ANOVA and the Sidak method of correction for pairwise multiple comparisons, or as specified in the figure legends. Normal and equal distribution of variances was assumed. Values were considered significant at $P < 0.05$. All analyses were performed using Prism version 7.0 software (GraphPad) or custom python scripts.

**Reporting summary**. Further information on research design is available in the Nature Research Reporting Summary linked to this article.

## Data availability
scRNA-seq data that support the findings of this study are deposited in the National Center for Biotechnology Information Gene Expression Omnibus (GEO) under the accession code GSE161125. All other data supporting the findings of this study are available with the article and its supplementary information files, and from the corresponding author upon reasonable request. Source data are provided with this paper.

## Code availability
The code used for processing and analyzing the data in this study is available in the following public GitHub repository: [https://github.com/Miller-JensenLab/munoz-rojas_NatCommunications2020]. Additionally, the custom MATLAB software for extracting and processing single-cell secretion data is available in the following public GitHub repository: [https://github.com/Miller-JensenLab/Single-Cell-Analysis].

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

## Acknowledgements

We thank Ruslan Medzhitov and Andre Levchenko for their insightful comments and discussion, as well as all members of the Miller-Jensen lab for helpful discussion and experimental advice. This work was supported by the National Institutes of Health (R01-GM123011 to K.M.-J.).

## Author contributions

Conception: A.R.M.-R. and K.M.-J. Development of methodology: A.R.M.-R. and K.M.-J. Acquisition of data: A.R.M.-R., I.K., J.P., and M.C. Analysis and interpretation of data: A.R.M.-R., I.K., and K.M.-J. Writing, review, and/or revision of the manuscript: A.R.M.-R., I.K., and K.M.-J. Study supervision: K.M.-J.

## Competing interests

The authors declare no competing interests.
