## [Peer Review File · Nature Communications]

Reviewer #1 (Remarks to the Author):

In this manuscript, the authors present important new scRNAseq data that points to (i) underappreciated heterogeneity in murine macrophage responses to cardinal polarization stimuli, and (ii) the presence of 'commitment' to cell stimulation that does not change (much) over time. This is the first of what will be many studies of single macrophage cell responses to diverse stimuli, which will reinforce the concept (long-held by leaders in the area) that macrophage activation is incredibly diverse, because that is what the biology of diverse inflammation and tissue repair demands. In essence, considerable chaotic process drive (temporal) resolving vs. non-resolving inflammation. Overall, I strongly support the data and concepts presented in this manuscript. There are some small alterations that may enhance the context:

1. The time kinetic (i.e. 6 hr., line 85) is a significant limitation of the study. The authors omit to mention that Arg1 is not an M2 marker and is in fact strongly induced by M1 activation signals through the autocrine-paracrine cytokine mechanisms articulated before (El Kasmi et al. Nature Immunol, Qualls et al. Science Signaling). The autocrine-paracrine pathways and their consequences need to be discussed in the context of the limitations of the study.
2. A second omission concerns the known diversity of BM-derived populations generated by GM-CSF (not used here, but should be mentioned); see Reis e Souza in Immunity and the subsequent discussion with Manfred Lutz and others. The fact is that the authors (and everyone else) are dealing with highly complex cell populations that are often assumed to be homogeneous (by most people).

Response to Reviewers

Overview

We are thankful to the reviewers for their thoughtful suggestions to improve our manuscript. We appreciate the reviewers overall positive comments, noting that “the authors present important new scRNAseq data”, and that “This is a characteristically insightful and thorough paper, and adds substantial insight to the debate surrounding M1-M2 polarization”. Both reviewers pointed out the novelty and significance of the experimental approach and computational methods, saying “this is the first of what will be many studies of single macrophage cell responses to diverse stimuli” and “I find this to be novel, exciting, and of broad interest”. Of note, both reviewers also fully supported the manuscript, indicating that they “strongly support the data and concepts presented in this manuscript” and “this paper is a great candidate for Nature Communications”.

The reviewers also offered several suggestions that further improved the manuscript. In summary, we made the following changes:

- In response to Reviewer 2’s questions regarding the neural network classifier, we revised our machine learning approach and we added UMAP visualization of co-stimulated macrophage heterogeneity to aid interpretation of our results (revised Fig. 3 and new Supp. Fig. 3).
- In response to Reviewer 1’s comments regarding the specificity of Arg1 as an M2 marker, we added additional data to show that Arg1 is specifically induced by IL-4 (and not LPS+IFN) through 24 hours in our experimental system (Supp. Fig. 2a); and we also added text to the Discussion to address how autocrine and paracrine signals might change that specificity in response to other M1 cues.
- In response to comments raised by both reviewers, we added a paragraph in the Discussion that addresses some of the limitations of our study that offer opportunities for future work.

We have provided responses the reviewer’s comments in blue below. The changes to the text have been highlighted in the revised manuscript.

Reviewer #1

In this manuscript, the authors present important new scRNAseq data that points to (i) underappreciated heterogeneity in murine macrophage responses to cardinal polarization stimuli, and (ii) the presence of ‘commitment’ to cell stimulation that does not change (much) over time. This is the first of what will be many studies of single macrophage cell responses to diverse stimuli, which will reinforce the concept (long-held by leaders in the area) that macrophage activation is incredibly diverse, because that is what the biology of diverse inflammation and tissue repair demands. In essence, considerable chaotic process drive (temporal) resolving vs. non-resolving inflammation. Overall, I strongly support the data and concepts presented in this manuscript. There are some small alterations that may enhance the context:

We thank the reviewer for these positive comments and support of the manuscript.

1. The time kinetic (i.e. 6 hr., line 85) is a significant limitation of the study. The authors omit to mention that Arg1 is not an M2 marker and is in fact strongly induced by M1 activation signals through the autocrine-paracrine cytokine mechanisms articulated before (El Kasmi et al. Nature Immunol, Qualls et al. Science Signaling). The autocrine-paracrine pathways and their consequences need to be discussed in the context of the limitations of the study.

We thank the reviewer for bringing up these important points. Regarding the kinetics of the response, we agree that performing scRNA-seq analysis at only one time point is a limitation of the study. Although six hours was carefully chosen as a compromise in terms of the different dynamics of the LPS+IFN vs. IL-4 response (as explored in Supp. Fig. 1a), it is likely that different aspects of response heterogeneity would be revealed by looking at shorter and longer time points (a point now added to the Discussion, see highlighted section on p. 17, line 359). For the finding of orthogonal regulation of *Ii6* and *Ii12b* vs. *Arg1* and *Chil3*, we note that the single-cell secretion data collected at 48 hours confirms that this aspect of response heterogeneity is durable over a longer time frame.

The reviewer raises an interesting point regarding the specificity of Arg1 as an M2 marker. In our experimental set-up, Arg1 is not induced by LPS+IFN γ , even at later time points. To demonstrate this, we updated Supp. Fig 2a with Arg1 protein expression as measured by flow cytometry showing that Arg1 is specific to IL-4 as late as 24 hours. This is clearly different from the cited manuscripts demonstrating Arg1 induction following mycobacteria infection. Given this variation across M1 stimuli—and to acknowledge the fact that many stimuli considered to be “M1” lead to different response profiles—we have generally restricted our labeling to be stimulus-specific, i.e., LPS+IFN-stimulated vs. IL-4-stimulated, rather than referring to them as M1 and M2. The differential induction of Arg1 might be explained by the binding of different TLRs (TRL4 vs TRL2), the presence of IFN γ , or other factors. We also agree that paracrine and autocrine signaling might play an important role in regulating the heterogeneity to both single and combined stimuli, as highlighted by the studies cited by this reviewer, as well as our own (Xue et al, Science Signaling, 2015). We added text in the Discussion to raise a potential role for autocrine and paracrine signaling in regulating a heterogeneous macrophage response (p. 18, line 375).

2. A second omission concerns the known diversity of BM-derived populations generated by GM-CSF (not used here, but should be mentioned); see Reis e Souza in Immunity and the subsequent discussion with Manfred Lutz and others. The fact is that the authors (and everyone else) are dealing with highly complex cell populations that are often assumed to be homogeneous (by most people).

We thank the reviewer for raising this important qualification about our study. We used M-CSF to differentiate monocytes to macrophages, while other studies have used GM-CSF. In both cases, the differentiated populations are complex and heterogeneous, as noted by the reviewer, and this will affect the heterogeneity observed in response to

subsequent cues, which we are analyze in this study. To address this point, we have added a section in the discussion that points out that our findings are specific to BMDMs differentiated with M-CSF, and differentiation with GM-CSF would likely lead to different sources of heterogeneity, such as the presence of dendritic cells and macrophages in the same population (p. 17, line 364).

Reviewer #2

In this exciting new manuscript, Miller-Jensen and colleagues probe the fluidity of M1-M2 macrophage polarization, recognizing that in a real environment individual macrophages will be subject to complex combinations of the M1 and M2 stimuli. Thus, they expose BMDMs to LPS and IFN γ (M1) as well as IL-4 (M2) to see how the cells will respond. They employ a fairly comprehensive approach, involving single cell analysis and advanced computational methods. This is a characteristically insightful and thorough paper, and adds substantial insight to the debate surrounding M1-M2 polarization. I have a few thoughts to add or ask about, but I think this paper is a great candidate for Nature Communications.

Thank you for the positive comments and support of our manuscript.

Our lab has also studied interactive effects in individual cells (at the risk of blowing my cover here), with LPS in combination with TNF. One thing that we found was that the concentrations mattered a lot - not just the concentration of each component, but also the concentrations relative to each other (Gutschow et al MBoC 2019). If the relative contribution of each stimulus was off, one stimulus could dominate the effect. I appreciated the work that the authors did to find a good combination of concentrations, but I think it's worth noting the fact that there are potentially many other combinations that could have very different effects. Another thing we found was the importance of timing, and in particular for us at least, how the extra information we obtained by considering multiple timepoints helped us to gain insight. Again, I appreciated the authors efforts to find a "best" time point, and I'm not necessarily recommending more time points for this study - but I think it's really important to at least note that following up this work with more timepoints could lead to significant new insights - in our own experience that may even seem to contradict or complicate some of the conclusions one might draw from the single time point. For example, Nos2 and Arg1 are known to compete for arginine, but how long it takes to see the negative cross-regulation is unclear. The "Mixed" population may be specific to an earlier time-point and may disappear at 48 h when secretion was measured.

Thank you for recommending that we directly address the fact that there will almost certainly be time-dependent and dose-dependent effects on heterogeneity that were not explored in our study. Reviewer 1 also pointed out the limitation of looking at a single time point. To address these issues, we have added a paragraph to the discussion to emphasize the many dimensions affecting such complex responses (p. 17, line 359). We expect this study will be one of many exploring so-called M1 and M2 complexity in

particular. Indeed, we are currently working on a follow-up study that seeks to understand how the duration and concentration of each stimulus affects the response of macrophages after single and co-stimulation.

With regard to “Co-stimulation with LPS+IFN- γ and IL-4 induces transcriptional cross-regulation that varies substantially across individual cells”, covered in Figs 2 and 3, the paper that I mentioned above also made a similar (and to me, surprising) finding - that there were fractions of the individual cells that activated one pathway or the other, but actually also a significant fraction that activated both (our Fig 3). It’s wonderful to see an independent lab and set of methods arrive at the same conclusion, and makes me think it must be something more general. Exciting to see.

Thank you for suggesting that we incorporate this additional context to our work—namely, that single-cell studies in macrophages exposed to complex cues have consistently revealed that individual cells can elicit distinct responses. We have added this point and additional references to the discussion (p. 17, line 373).

One concern from this part of the manuscript relates to the machine learning in Figure 3. The results don't seem to be robust to the number of genes or types of genes used to do the classification, and the difference between the two panels in Fig. 3b suggests the overfitting of the model. Some suggestions to help this would be: to incorporate model validation (showing MSE from the cross validation) or try a simpler classifier that was able to classify the cells based on a larger subset of the unique core genes. Alternatively, if they did UMAP (or other dimension reduction) with the co-stimulated cells only and got clusters of similar proportion to the machine learning results that would offer evidence supporting the machine learning results as being representative of the underlying structure of the data.

Thank you for pointing out that the manuscript would benefit from further analysis to explain the results from the classifier. To clarify, for each set of genes we built a unique neural network that was trained using only that specific set of genes. The motivation for this analysis was to explore how cell classification depended on the set of genes being considered. For each classifier, we performed cross-validation with a reserved subset of the single-stimuli data, and all classifiers had a >99% accuracy when classifying single-stimulated cells. We appreciate that you pointed out that we had neglected to include this, and we have added the relevant text to the methods section (p. 24, line 510).

In addition to these technical clarifications, your comments caused us to broadly rethink the entire analysis (p. 9, starting on line 188). Per your suggestion, we performed UMAP analysis to visualize clusters of co-stimulated macrophages. To aid interpretation, we calculated a combined gene-expression score for the unique core genes (UCGs) induced by LPS+IFN- γ or IL-4 for each single cell (i.e., a metric to capture the extent to which exhibited “LPS+IFN- γ -like” or “IL-4-like” expression). When we overlaid these scores on the UMAP plot, we observed a clear separation between the cells that had a high LPS+IFN- γ UCG score and the cells that had a high IL-4 UCG score, suggesting selective expression of these gene sets by co-stimulated cells (new **Fig. 3a**), and motivating the use of a machine-learning approach.

For the NN classifier, we substantially simplified our approach by training LPS+IFN- γ and IL-4 classifiers on one of two gene signatures: 1) all genes or 2) UCGs downregulated by co-stimulation. Classification based on all genes predicted a substantial “Mixed” cell population; however when we restricted our training set to UCGs negatively regulated by co-stimulation, the number of co-stimulated cells classified as “Mixed” was reduced and a fraction of cells exhibiting an IL-4-dominant signature emerged (revised **Fig. 3c**). In fact, the fractions of co-stimulated cells classified as “LPS+IFN- γ -dominant” or “IL-4-dominant” were both substantially larger than the “Mixed” and “Unclassified” fractions when trained on this gene set. When plotting the classification results back onto the UMAP representation, the cells classified as “LPS+IFN- γ -dominant” or “IL-4-dominant” were the same cells that had high LPS+IFN- γ or IL-4 UCG scores, respectively (new **Supplementary Fig. 3**). Overall, we appreciate your suggestions, as we think this revised analysis is much easier to interpret and more clearly illustrates the response heterogeneity within the co-stimulated population.

I was also very interested in the results of Fig 2e, in particular the identification of Saa3. In another paper we were able to connect single cell gene expression and NF- κ B activation in the same cell (Lane et al Cell Systems 2017), and we also identified Saa3 in what I thought was the most interesting part of that study (Fig 3). Saa3 is only expressed in individual cells that exhibit an oscillatory-like pattern of NF- κ B - not a stable activation pattern, a transient pulse or no activation. It was the strongest signal from that subset of cells. Ever since then I’ve been wondering about Saa3, so to find it identified in this study was really interesting.

Another observation that was left on the cutting floor for that paper: it seemed to us at the time that there were certain pairs of genes that were usually or always expressed in the same cell, and other pairs that rarely or never were expressed in the same cell. The statistics of single cell RNA seq didn’t allow us to go farther with that, so we had to leave it out - but this manuscript really establishes it (starting at Line 206) and it’s terrific - I find this to be novel, exciting, and of broad interest. Tying it to a mechanism via the identification of Nfkbiz and Klf4 is icing on the cake.

Thank you for your enthusiasm about our study. We were also really excited to see how efficiently calculating the log(odds ratio) for expression of transcripts across individual cells revealed potential “modules” of expression that were distinct across cells, including the TFs that might regulate those modules. We’re now following up to experimentally test these biological mechanisms.

Thank you also for raising the connections to Keara Lane’s manuscript. Although we did not explore heterogeneity within the cells stimulated with LPS+IFN- γ alone, it clearly exists and supports the observations you describe. Specifically, Saa3 and Il6 (as well as Il12b) have a significant positive log(odds ratio) following LPS+IFN treatment indicating that they are produced together. This is consistent with the fact that Saa3 and Il6 were both differentially expressed in response to the same “Recurrent” oscillatory NF- κ B signaling pattern identified in subpopulation 3 in the Lane study. An interesting future

direction would be to assess if stimulation with IL-4 directly impacts NF-kB signaling dynamics via cross-regulation, or if IL-4 instead changes how downstream genes respond to the same dynamics.

Given the finding in lines 98-100 that many more genes are downregulated, it might make sense to consider a few of these genes in more detail. Most of the paper is about upregulated genes.

We acknowledge that the downregulated genes remain largely unexplored. We prepared tables of all the genes significantly upregulated and downregulated in response to LPS+IFN and IL-4 relative to control (**Supplementary Data 1** that accompanies **Fig. 1b**) with the expectation that other labs might find interesting results in those data sets. We will make the entire data set publicly available upon publication.

REVIEWERS' COMMENTS

Reviewer #1 (Remarks to the Author):

The minor comments have been addressed

Reviewer #2 (Remarks to the Author):

The authors have resolved my concerns, and I congratulate them on their fine work.

Response to Reviewer's Comments

Reviewer #1 (Remarks to the Author):

The minor comments have been addressed

We thank the reviewer for their comments.

Reviewer #2 (Remarks to the Author):

The authors have resolved my concerns, and I congratulate them on their fine work.

We thank the reviewer for his comments and kind words.